# A Novel Defined PANoptosis-Related miRNA Signature for Predicting the Prognosis and Immune Characteristics in Clear Cell Renal Cell Carcinoma: A miRNA Signature for the Prognosis of ccRCC

**DOI:** 10.3390/ijms24119392

**Published:** 2023-05-28

**Authors:** Yanmei Wang, Jia Zhou, Nan Zhang, Yiran Zhu, Yiming Zhong, Zhuo Wang, Hongchuan Jin, Xian Wang

**Affiliations:** 1School of Medicine, Zhejiang University, Hangzhou 310030, China; wangyanmei@zju.edu.cn (Y.W.); 18868024662@163.com (N.Z.); 11718259@zju.edu.cn (Y.Z.); zhongyiming@zju.edu.cn (Y.Z.); drzhuowang@zju.edu.cn (Z.W.); 2Laboratory of Cancer Biology, Key Lab of Biotherapy in Zhejiang Province, Cancer Center of Zhejiang University, School of Medicine, Zhejiang University, Hangzhou 310016, China; jinhc@zju.edu.cn

**Keywords:** PANoptosis, microRNA, clear cell renal cell carcinoma, prognostic signature, tumor microenvironment

## Abstract

Clear cell renal cell carcinoma (ccRCC) is one of the most prevalent cancers, and PANoptosis is a distinct, inflammatory-programmed cell death regulated by the PANoptosome. The essential regulators of cancer occurrence and progression are microRNAs (miRNAs). However, the potential function of PANoptosis-related microRNAs (PRMs) in ccRCC remains obscure. This study retrieved ccRCC samples from The Cancer Genome Atlas database and three Gene Expression Omnibus datasets. PRMs were recognized based on previous reports in the scientific literature. Regression analyses were used to identify the prognosis PRMs and construct a PANoptosis-related miRNA prognostic signature based on the risk score. We discovered that high-risk patients had poorer survival prognoses and were significantly linked to high-grade and advanced-stage tumors, using a variety of R software packages and web analysis tools. Furthermore, we demonstrated that the low-risk group had significant changes in their metabolic pathways. In contrast, the high-risk group was characterized by high immune cell infiltration, immune checkpoint expression, and low half-maximum inhibition concentration (IC50) values of chemotherapeutic agents. This suggests that high-risk patients may benefit more from immunotherapy and chemotherapy. In conclusion, we constructed a PANoptosis-related microRNA signature and revealed its potential significance in clinicopathological features and tumor immunity, thereby providing new precise treatment strategies.

## 1. Introduction

Kidney and renal pelvis cancer caused 76,080 new cases and 13,780 deaths in the United States in 2021, accounting for 4% of all new cancer cases. In addition, its annual incidence is gradually rising [1]. Clear cell renal cell carcinoma (ccRCC) is kidney cancer’s most prevalent histological subtype [2]. Advanced ccRCC has high morbidity and mortality rates due to the disease’s high level of malignancy [3]. A physical resection or ablation is typically used to treat localized cancer. Antiangiogenic therapy has improved survival significantly as the standard first-line treatment for metastatic renal cell carcinoma (mRCC) [4,5]. In addition, immune checkpoint inhibition (ICI) has also been demonstrated to be an effective new clinical management strategy. Due to the immunogenicity of mRCC, researchers have studied combinations of angiogenesis-targeted therapy and ICIs. According to a relative network meta-analysis, pembrolizumab plus axitinib are likely the preferred agents for progression-free survival (PFS) and overall survival (OS) in mRCC [6]. However, due to the high heterogeneity of renal cell carcinoma, its treatment is still limited, and individual precision cannot be achieved to improve its clinical outcomes. Therefore, establishing a reliable new prognosis signature will provide an accurate treatment direction.

When cells are stimulated by internal and external environmental factors, programmed cell death (PCD) is triggered to eliminate pathogens and maintain cellular homeostasis [7]. Among all the PCD pathways, apoptosis, pyroptosis, and necroptosis are the three major pathways that have been studied the most thoroughly. Apoptosis is a program that forms apoptotic bodies and avoids inflammatory responses; pyroptosis and necroptosis rupture cell membranes and release inflammatory factors to induce inflammatory responses [8]. A growing number of studies have discovered that not only are their internal mechanisms complex, but they also interact with one another. For example, ZBP1 acts as a sensor of the influenza A virus. It promotes interaction between RIPK3, RIPK1, caspase-8, caspase-6, and the NLRP3 inflammasome through homotypic or heterotypic domains, forming a complex to simultaneously activate apoptosis, pyroptosis, and necroptosis in macrophages [9]. PANoptosis is defined as an inflammatory PCD pathway that is activated by specific triggers and regulated by the PANoptosome complex, with pyroptosis, apoptosis, or necroptosis characteristics that cannot be explained by these PCD pathways alone [8,10]. Recent research has demonstrated that the activation of PANoptosis is a response to viral, bacterial, and fungal triggers and is associated with autoimmune diseases, cytokine storms, and cancer [11]. Therefore, inducing and regulating inflammatory cell death has been considered as a cancer treatment. The Food and Drug Administration (FDA) has approved many drugs that can induce this inflammatory cell death, such as doxorubicin and oxaliplatin. However, they lack pathway targeting and cause numerous side effects [12]. The comprehensive properties of PANoptosis mechanisms provide a thorough understanding of inflammatory cell death, which is a new research hotspot, given the inflammatory nature of PANoptosis. Therefore, further research on the molecular regulation of PANoptosis in cancer models is urgently required to identify new molecular targets for anticancer immunotherapy.

MicroRNAs (miRNAs) are ~22 nt small noncoding RNAs that play a crucial role in the post-transcriptional regulation of messenger RNA (mRNA). miRNAs inhibit expression by binding to the 3’UTR of their target mRNAs [13]. The literature extensively reports that miRNAs are widely dysregulated in human cancers, highlighting their crucial roles in tumorigenesis, growth, and metastasis [14]. A comprehensive understanding of the role of miRNAs in cancer has made them attractive tools and targets for new therapeutic approaches. Giovanni et al. developed an algorithm based on the urinary levels of miR-122-5p, miR-1271-5p, and miR-15b-5p, as well as three controls, to aid in the early diagnosis of ccRCC, demonstrating the clinical utility of miRNAs [15]. Although numerous articles have investigated the regulation of miRNAs in the respective pathways of apoptosis, pyroptosis, and necroptosis, none of the literature systematically reports the research on PANoptosis and its related miRNAs. Exploring the relationship between PANoptosis-related microRNAs (PRMs), the survival prognosis, and the potential characteristics of ccRCC patients will provide a new clue for developing individualized treatment plans.

To address these issues, this study downloaded the miRNA expression, mRNA, and valid clinical data of ccRCC patients from public databases. Concurrently, we sorted through the literature to extract information about PANoptosis-related miRNAs. Then, a prognostic multi-miRNA signature model was created using differential and regression model analyses. Finally, the clinical characteristics, tumor immunity, and potential therapy mechanisms were thoroughly analyzed, shedding light on incorporating prognostic prediction into precision therapy.

## 2. Results

### 2.1. Identification of Differentially Expressed miRNAs between Normal and Tumor Tissues

In order to introduce the entire research process, we first summarized a flowchart (Figure 1). The first step was to develop a PANoptosis-related miRNA prognostic signature model, followed by a discussion of its clinical characteristics and an investigation of its immune infiltrate characteristics and therapeutic studies. The TCGA database contained the mRNA and miRNA expression data of 71 normal kidney tissues and 545 kidney tumors, as well as their corresponding clinical variables and mutation data. Previous literature reports identified 11 miRNAs closely related to PANoptosis (Appendix A) [16,17,18,19,20,21,22,23,24,25,26,27,28,29,30,31,32,33,34,35,36,37,38,39,40,41,42,43,44,45,46,47,48,49,50,51,52,53,54,55,56,57,58,59,60,61,62,63,64,65,66]. However, the TCGA database’s differential analysis between normal and cancerous tissue revealed nine distinct miRNAs that merited further investigation (Figure 2A, Appendix A). In the tumor group, seven miRNAs (hsa-miR-155-5p, hsa-miR-15a-5p, hsa-miR-16-5p, hsa-miR-181a-5p, hsa-miR-21-5p, hsa-miR-210-3p, and hsa-miR-223-3p) were upregulated, while two miRNAs (hsa-miR-141-3p and hsa-miR-200a-5p) were downregulated. Figure 2B depicts the correlation network that contains these miRNAs.

### 2.2. Development of the PANoptosis-Related Prognostic Signature Model

We eliminated the cancer patient samples with incomplete clinical information and were left with 511 valid patient samples. We randomly divided the patients into a training group (*n* = 256) and a testing group (*n* = 255). The testing group could be used for model validation in the future. Table 1 displays the fundamental files. We performed a univariate Cox regression analysis on the patients in the training group, preliminarily screened the miRNAs that significantly correlated with the patients’ survival times, and obtained four candidate miRNAs that met the *p* < 0.05 criterion (Figure 2C). Three miRNAs were then chosen as the best scoring system. Their regression coefficients were calculated using a least absolute shrinkage and selection operator (Lasso) Cox regression analysis to reduce the number of candidate genes and establish a signature (Figure 2D,E). Each sample’s risk score was calculated as follows: Risk Score = (−0.5810 × hsa-miR-200a-5p expression) + (0.9575 × hsa-miR-21-5p expression) + (0.9217 × hsa-miR-223-3p expression) (Appendix A). The patients’ prognoses worsened with higher expressions of hsa-miR-21-5p and hsa-miR-223-3p, while the patients’ prognoses improved with higher expressions of hsa-miR-200a-5p (Figure 2F–H).

### 2.3. Assessment of the PANoptosis-Related Prognostic Signature Model

The 256 patients in the training group were divided into high- and low-risk subgroups based on the median risk score (Figure 3A, left panel). We discovered that patients in the high-risk group had a higher mortality rate (Figure 3B, left panel). According to the principal component analysis (PCA) plot, patients with various risks were divided into two clusters (Figure 3C, left panel). The expression of three OS-related PRMs between the high- and low-risk groups in the training group was presented as heatmaps (Figure 3D, left panel). The difference in OS made it clear that the survival prognosis for the high-risk patients was significantly worse (Figure 3E, left panel). Lastly, a time-dependent receiver operating characteristic (ROC) curve was utilized to evaluate the sensitivity and specificity of the 3-miRNA signature model. The area under the ROC curve (AUC) values were 0.734 at one year, 0.721 at three years, and 0.718 at five years, indicating that the model had a good predictive ability (Figure 3F, left panel). We applied the risk score formula to 255 patients in the testing group (Figure 3A–F, middle panel) and the entire group (Figure 3A–F, right panel). We came to the same conclusions as above, validating the practicability and reliability of this risk prognosis model.

### 2.4. Clinical Characteristics of the Risk Score Model

A heatmap was used to compare the distribution of the clinicopathological characteristics between the risk groups and investigate the relationship between the risk score and each clinicopathological characteristic (Figure 4A). According to the chi-square test, high-grade and advanced-stage tumors were significantly associated with the high-risk group, whereas low-grade and early-stage tumors were associated with the low-risk group (Figure 4B–E). These findings indicated that the PANoptosis-related risk score might play a crucial role in the progression of ccRCC tumors. To determine whether the risk score was generally applicable to patients with various clinical characteristics, we performed a stratified survival analysis according to the following clinical characteristics: age (≤65 or >65); gender (male or female); grade (G1–G2 or G3–G4); and AJCC stage (I–II or III–IV). According to the survival analysis, the OS was significantly worse for the high-risk group than for the low-risk group, demonstrating that consistent results were obtained in each subgroup (*p* < 0.05) (Figure 4F–I). In addition, we combined the risk score with clinical factors for a univariate Cox regression analysis and multivariate Cox regression analysis, in order to assess whether the risk score could become an independent prognostic factor. We evaluated the hazard ratio (HR) and 95% confidence interval (CI) in univariate and multivariate Cox regression models. Finally, we discovered that grade (HR = 1.4002, 95%CI: 1.1040–1.7761, *p* = 0.0055) and risk score (HR = 1.5411, 95%CI: 1.3362–1.7776, *p* < 0.001) were the independent prognostic factors based on the multivariate analysis results (Figure 5A,B). We also performed ROC analyses on the clinical variables and discovered that the risk score had an excellent predictive performance (Figure 5C). We developed a nomogram to predict the one-year, three-year, and five-year overall survival of ccRCC patients based on the above independent prognostic indicators (Figure 5D). The calibration curve also demonstrated that the predicted survival times at one, three, and five years were comparable with the reference line, indicating that the nomogram was an ideal prediction model (Figure 5E).

### 2.5. The Relationship between Metabolic Process and Risk Groups

To further investigate the functional differences between the high- and low-risk groups, gene set enrichment analysis (GSEA) on a gene ontology (GO) analysis and Kyoto Encyclopedia of Genes and Genomes (KEGG) analysis revealed that the genes in the high-risk group were predominantly enriched in the following content: chronic inflammatory response, cytokine activity, and granulocyte chemotaxis (Figure 6A). Notably, numerous metabolic pathways were also observed in the low-risk group, such as the amino acid betaine metabolic pathway and the fatty acid catabolic process (Figure 6A,B). This demonstrates how these metabolic processes may have an impact on the survival prognoses of patients. Metabolic alterations are a common characteristic of most cancer cells, and this is well known. According to the concentrations and compositions of nutrients in their microenvironment, tumor cells can independently select the optimal energy supply mode for their growth, influencing the biosynthesis capacity of fatty acids, amino acids, and ATP [67,68]. Therefore, we further investigated the relationship between metabolic-related genes and patients. We identified 137 differentially expressed genes (DEGs) between the high- and low-risk groups. Through GSEA-related pathways, 118 genes related to amino acid metabolism and 150 genes related to lipid metabolism were collected, and the above genes intersected with the differentially expressed genes (Figure 6C). The results showed that two genes (MFSD2A and CYP24A1) were positively correlated with the risk scores and three genes (CYP4A11, PPARGC1A, and PCK1) were negatively correlated with the risk scores (Figure 6D–H). Among them, the PCK1 gene was involved in lipid and amino acid metabolism, indicating its potential importance in the metabolic regulation of renal cancer patients.

### 2.6. Characteristic Performance of the Tumor Microenvironment Cell Infiltration

The above GSEA results revealed that the high-risk group had an enrichment of the immune pathways previously associated with PANoptosis [69]. We calculated the stromal score, immune score, estimate score, and tumor purity using the estimate algorithm to compare the immune microenvironments between the low-risk and high-risk groups. The higher the ESTIMATE score, the higher the non-tumor component, and the lower the tumor purity. The findings demonstrated that the immune and stromal components of the high-risk group were greater than those of the low-risk group (Figure 7A–D), indicating that the immune states of the patients in the high-risk group were significantly activated. To investigate the immune cell infiltration in the high- and low-risk groups, various software, including XCELL, TIMER, QUANTISEQ, MCPCOUNTER, EPIC, CIBERSORT-ABS, and CIBERSORT, were used to study their microenvironments. The analysis results were presented as a bubble chart, illustrating the relationship between the risk score and immune cell infiltration, with CD8+ T cells being positively correlated with this risk score (Appendix A). In total, 22 types of immune cells identified by the CIBERSORT analytical tool were displayed (Appendix A), and a difference analysis for a fraction of immune cells revealed significant differences in 7 immune cells. In the high-risk group, CD4 memory T cells and M0 macrophages were significantly activated (Figure 7E). We also compared the relationship between cell infiltration and OS and found that a high infiltration of these CD4 memory T cells or M0 macrophages was associated with poor prognoses (Appendix A). Using an ssGSEA, we compared the enrichment scores of 16 immune cells and the activities of 13 immune-related pathways in the groups at a high and low immune risk. The findings indicated that the immune state in the high-risk group was more active, as indicated by APC co-stimulation and T cell co-stimulation, in addition to the fact that the high-risk group generally had more immune cell infiltration, such as CD8+ T cells, dendritic cells (DC) cells, and macrophages (Figure 7F,G). The expression levels of major histocompatibility complexes (MHC) tended to be higher in the high-risk group (Figure 7H), indicating a significant association between the high-risk group and stable antigen-presenting function. Significantly, T-cell regulatory (Treg) cells, which play a negative regulatory role in immunity, and cancer-associated fibroblasts (CAFs), which provide a good barrier environment for the development of tumors [70], were expressed in the high-risk cell groups (Figure 7E,F), where immune checkpoints were also activated (Figure 7G), indicating that immunosuppression also existed in these high-risk cell groups.

Single-cell sequencing methodologies are being increasingly used to reveal gene expression status at the cellular level [71] and provide a comprehensive description of genetic complexity [72], enabling us to better understand the heterogeneity and microenvironment of tumors. In order to learn more about the immune microenvironment, we analyzed the single-cell RNA-seq (scRNA-seq) data from three ccRCC samples from GSE203612. Each sample’s gene number and sequencing depth were displayed (Figure 8A,B). The content of mitochondrial genes, the traditional quality control index of a single-cell analysis, was evaluated, and it showed an excellent cell viability (Figure 8C). The sequencing depth was positively correlated with the mitochondrial content (Figure 8D) and intracellular sequences (Figure 8E). The volcano plot depicted fluctuating genes in all the samples, and we screened the 1500 genes with the most pronounced fluctuations for further study (Figure 8F). A PCA was used to eliminate the dimensions with significant separation (Figure 8G), and the tSNE algorithm was used to reduce the dimensions of the top 15 PCs (*p* < 0.0001) to produce 11 clusters (Figure 8H). The heatmap depicted the marker genes in each cluster (Figure 8I). The clusters were then annotated into eight types of cells based on their marker genes, revealing an abundant immune cell infiltration in the ccRCC samples (Figure 8J). The results showed that macrophages and T cells comprised most of the immune cells in the microenvironment, which was consistent with previous CIBERSORT data (Appendix A). Then, using single-cell trajectory and pseudotime analyses, we obtained three branches in differentiated cells that changed over pseudotime (Figure 8K) and could infer the differentiation process of the cells in the TME (Figure 8L,M). It was determined that monocytes first differentiated into macrophages, then T cells and NK cells rapidly increased. The accumulation of tumor epithelial cells and endothelial cells led to tumor development.

### 2.7. Estimation of Immunotherapy and Chemotherapy Responses

To determine whether the risk score had a significant correlation with the immunotherapy effect, we next checked a series of evaluating indicators, including the IPS score, TMB score, and immune checkpoint molecules. This was performed considering the above studies’ clear correlation between the risk groups and the immune microenvironment. First, the IPS scores were used to evaluate the immune system activation of immunotherapy in various risk groups [73]. The high-risk group had higher scores for antigen presentation and effector cells and lower scores for suppressor cells. However, the immune checkpoint was slightly different from the previous prediction results, and there was no significant difference in the IPS comprehensive score (Figure 9A). TMB is a novel method for calculating tumor cells’ mutations that can be used as a biomarker to evaluate the efficacy of immunotherapy for cancer [74]. We applied the PANoptosis-related signature to the TMB calculation, and mutation landscapes of the top 20 most mutated genes, with VHL and PBRM1 making up the majority, were displayed in the waterfall (Figure 9B,C). The missense mutation form of VHL was more prevalent in the high-risk group, whereas the nonsense mutation of PBRM1 was more prevalent in the low-risk group. In addition, the mutation rate of BAP1 was significantly higher in the high-risk group (Figure 9D). The high-TMB sub-group had a worse prognosis for survival than the low-TMB sub-group, according to an additional survival analysis (Figure 9E). The sub-group with high-risk scores and a high TMB had the lowest survival probability, whereas the sub-group with either a low-risk score or a low TMB had better survival benefits (Figure 9F). TMB reflects the immunogenicity of a tumor, so patients with a high TMB have low survival rates in many cancers [75]. However, it is also an opportunity for patients with a high TMB and risk score to benefit from immunotherapy. In addition, we compared the expressions of the immune checkpoint inhibitor targets between the two groups. According to the results, five major immune checkpoint molecules (CTLA4, LAG3, PD-1, SIGLEC15, and TIGIT) were significantly upregulated in the high-risk group (Figure 9G). This indicated that the patients in the high-risk group might have been more sensitive and effective in applying immune checkpoint inhibitors. 

Next, we evaluated the susceptibilities of the low- and high-risk populations to anticancer drugs. Except for vinorelbine, common broad-spectrum chemotherapy drugs, such as camptothecin, cisplatin, and docetaxel, were generally more effective in the high-risk patients (Figure 10A–I). The low half-maximum inhibition concentration (IC50) values of sunitinib, temsirolimus, and pazopanib, three targeted chemotherapeutic agents used clinically to treat advanced renal cancer, were lower in the patients with high-risk scores (Figure 10J–L). A renal cell carcinoma dataset (GSE74174) of 74 ccRCC patients treated with tyrosine kinase inhibitor (TKI) was chosen to predict the responses to TKI therapy. The patients were divided into two groups based on their responses to TKI treatment: progressive disease (PD); stable disease (SD), partial response (PR), or complete response (CR), collectively referred to as non-PD. After normalizing the raw data, we applied the above risk score formula to the 74 patients to calculate their risk scores. Based on the median risk score, the patients were divided into high- and low-risk groups (Figure 10M and Appendix A). However, the findings indicated no statistically significant differences between the response types (PD and non-PD). Since this dataset only indicated PD or non-PD, the sample size was insufficient, and there were no additional specific time data. More data are required to support this treatment response study. These findings suggested that the high-risk group exhibited a better response to immunotherapy and a better effect with chemotherapeutic drugs, thereby providing potential treatment options for kidney cancer.

### 2.8. Functional Enrichment Analysis of Distinctive miRNAs

We used the GSE16441 microarray to confirm the expression levels of three miRNAs (Figure 11A and Appendix A), which matched the TCGA database characteristics. Clinical clear cell renal cell carcinoma samples and adjacent normal samples were used to confirm the expression levels of these three miRNAs (Figure 11B, Appendix A–C, and Appendix A). The detailed clinical characteristics of the patients are provided in Appendix A. In the tumor samples, miR-200a-5p was downregulated while miR-21-5p was upregulated, which is consistent with previous findings. However, there was no significant difference in the miR-223-3p expression, possibly due to significant individual differences and a lack of clinical samples to conclude. Next, we investigated the potential functional roles of microRNAs in renal cancer. We used the miRDB, TargetScan, and miRTarBase databases to predict the miRNA target genes (Appendix A). Ultimately, 120 genes were screened and the Cytoscape software (version 3.8.2) was used to establish the relationship between the miRNAs and mRNAs (Figure 11C). Among them, the miR-223-3p-targeted NLRP3 gene has been studied extensively, indicating its role as a vital receptor component of the inflammasome. It can induce inflammation and promote cell death in response to signal stimulation [76,77]. We performed a GO analysis and KEGG enrichment analysis on these genes (Appendix A) and the results indicated that the target genes were significantly correlated with the MAPK signaling pathway (Figure 11D,E), which is involved in numerous physiological and pathophysiological processes, including cell growth, proliferation, apoptosis, and other biological processes. Subsequently, we examined the effect of miRNAs on cell growth in ccRCC. We estimated the expression levels of the miRNAs in two ccRCC cell lines (Caki-1 and 786-O) and discovered that the expression levels of these miRNAs in 786-O were relatively low (Appendix A). The structural design of the miRNA inhibitor was presented (Appendix A) and the transfection efficiency of miRNA mimics in 786-O and Caki-1 cells was assessed (Figure 11F and Appendix A). Using the Cell Counting Kit-8 (CCK-8) assay, we discovered that an upregulated expression level of miR-21-5p or miR-223-3p significantly enhanced the ability of renal cancer cells to proliferate. In contrast, a down-regulated expression level inhibited this ability. The outcome of miR-200a-5p was contrary to the findings of miR-21-5p and miR-223-3p (Figure 11G, Appendix A). These results were consistent with the oncogenic or tumor-suppressive properties of the miRNAs.

## 3. Discussion

New evidence has suggested that PANoptosis plays a significant role in the occurrence and development of tumors. Due to the complexity of PANoptosis, however, there are currently no systematic studies elucidating its effects on the prognosis and tumor microenvironment of ccRCC. MiRNAs are excellent candidates for developing minimally invasive biomarkers for cancer diagnosis and prognosis due to their function in cancer, their relative stability, and their resistance to storage [78]. In addition, a valid diagnostic decision making algorithm can facilitate the management of disease surveillance [79]. Therefore, adding miRNAs into the diagnostic algorithm will improve progress detection and guide treatment selection, without altering the natural history of the disease. Consequently, we developed a PRM-based algorithmic model to predict the prognoses of KRIC patients and provide immune insights, which can serve as a foundation for future mechanism research and provide additional clinical treatment strategies.

In this study, 11 PRMs were gathered from the literature. Nine miRNAs were screened as candidate genes, using a differential analysis of normal and cancer renal tissue. To further assess the prognostic value of these relevant regulators, we developed a three-miRNA signature model using a univariate Cox regression analysis and Lasso Cox regression analysis. In this study, the risk scores accurately predicted the survival of the patients in either the training or testing groups. According to the univariate and multivariate Cox regression analyses, this risk score was an independent prognostic factor for ccRCC patients. Next, a stratified survival analysis of various clinical characteristics confirmed the model’s broad applicability. Additionally, a nomogram was constructed and validated based on the risk score and significant clinical features. Due to the high consistency between the nomogram prediction for the ccRCC patients and the actual 1-, 3-, and 5-year survival rates, this may provide clinicians with a useful prognostic tool for analysis. In addition, we discovered that inflammatory responses were enriched and correlated with the high-risk groups based on a GSEA analysis of the high- and low-risk groups. Subsequently, the analysis of immune cell infiltration and microenvironmental characteristics revealed that the infiltrating immune cells were generally higher in the high-risk group, and some immune-related pathways remained active. In the meantime, immunosuppression also existed, suggesting that treatment with immune checkpoint inhibitors would improve treatment outcomes. Amino acid metabolism and active manifestations of lipid metabolism were also observed in the low-risk cell group; this metabolic change may result from the creation of an immune microenvironment hostile to T cells. To investigate the potential pathway mechanism from multiple perspectives, we also established an miRNA–mRNA co-expression regulatory network.

PANoptosis, a unique inflammatory programmed cell death regulated by the PANoptosome, is essential for restricting a broad spectrum of pathogens (including bacteria, viruses, fungi, and parasites) [80]. As a result, there have been numerous reports on its association with infection, while there are currently few studies on tumors. For instance, IRF1 activates PANoptosis to prevent AOM/DSS-induced colorectal tumorigenesis, indicating a promising future for PANoptosis-related cancer research [81]. Our study established a signature model with three related miRNAs (hsa-miR-21-5p, hsa-miR-223-3p, and hsa-miR-200a-5p) that could be used to predict the OS in ccRCC patients. According to the previous literature, Andrographolide inhibits the expression of NF-κB, and as a result, reduces the expression of miR-21-5p, dramatically increasing the target gene expression of programmed cell death protein 4 (PDCD4) to induce apoptosis in breast cancer [82]. On the other hand, it was discovered that an overexpression of miR-21-5p downregulates TGFBI, resulting in pyroptosis and the release of inflammatory factors in colorectal cancer [48]. Su et al. found that DHT inhibits cell growth and induces apoptosis by suppressing the miR-200a-5p expression in HepG2 and HT-29 cells [83]. In addition, an overexpression of miR-200a-5p induces receptor-interacting serine/threonine kinase 3 (RIP3)-dependent necroptosis in cardiomyocytes. It has been discovered that miR-21-5p and miR-200a-5p inhibit apoptosis and promote pyroptosis or necroptosis, suggesting that inhibiting one pathway in apoptosis will result in the death of other pathways [84]. miR-223-3p was discovered to be the first human miRNA to target NLRP3 directly [85]. In myeloid cells, the miR-223-3p expression and NLRP3 mRNA levels were negatively correlated [86]. NLRP3 has been identified as one of the sensor molecules of inflammasomes, which are endogenous, multimeric protein complexes that mediate the cleavage of proinflammatory cytokines, particularly IL-1β and IL-18 [87]. A high expression of miR-223-3p was also associated with a poor survival prognosis and was identified as a risk factor in our study. We hypothesize that miR-223-3p regulates inflammatory cell death and negatively correlates with apoptosis.

In general, PANoptosis is characterized by a hydrolytically damaged cell membrane and the release of its contents, and it directly triggers inflammatory responses. Immune infiltration is closely related to the therapeutic responsiveness and prognoses of ccRCC patients [88]. After the GSEA results revealed significant immune pathway enrichment in the high-risk group, an in-depth analysis of the potential features was conducted. Our study revealed that the high-score group was characterized by high stromal cells, immune cell infiltration, and a low percentage of tumor purity. Then, predictions were made using a variety of software algorithms, such as CIBERSORT and ssGSEA, and even a single-cell sequencing analysis. The results demonstrated that the high-risk group had more immune cell infiltration, including CD8+ T cells, dendritic cells, and macrophages. Treg cells and the environment’s buildup of CAFs also had a counteracting effect. These characteristics indicated that the high-risk group manifested an immune-inflamed phenotype. Inflamed tumors can exhibit immune cell infiltration with various subtypes, including immune-inhibitory regulatory T cells, myeloid-derived suppressor cells, suppressor B cells, and cancer-associated fibroblasts [89]. The immune-inflamed phenotype is also characterized by increased IFN-γ signal activation, antigen presentation, and high levels of PD-L1 expression [90]. Immune cells are abundantly present in inflamed tumors, but their function is suppressed, making them more susceptible to immune checkpoint inhibitors.

The risk score was positively correlated with tumor progression and negatively correlated with survival prognosis. The high-risk group with a poor survival prognosis had immune cells with a high infiltration, indicating that the patients’ immune functions were further suppressed as the tumor grew. Immune evasion is a classic function of cancer cells, which confront and deceive the immune system [91]. Immune evasion includes tumor cell antigen modification or deletion [75], a low expression of MHC class I molecules in tumor cells, an absence of co-stimulatory signals, and tumor-induced immunosuppressive effects [92]. We observed, however, that the high-risk group had a high MHC expression, high co-stimulator activity, and stable TMB, which were not immune escape mechanisms. The expression of immune checkpoints (CTLA4, LAG3, PD-1, SIGLEC15, and TIGIT) was significantly high and regarded as the primary cause of immune escape. As predicted by the prior immune-inflamed phenotype, immune checkpoint inhibitors were more likely to exert antitumor effects in the high-risk group in our study.

According to the previous analysis, the high-risk group was classified as having an immune-inflamed phenotype characterized by innate immune cell infiltration and immune activation. Like the high-risk group, the low-risk group was considered to have a relatively low degree of immune infiltration due to its unique characteristics. It is understood that a lack of immunity alters the metabolism of the tumor microenvironment, creating a hostile environment for T cell function and proliferation [93]. Hypoxia, lactate production, the presence of an acidic tumor microenvironment, and an increase in lipogenesis alter the immune metabolism of T cells, affecting the participation of T cell receptors; T cells affect activation, differentiation, and proliferation, resulting in a decrease in tumor-infiltrating lymphocytes (TILs) [94]. In line with the previous findings, we discovered that fatty acid and amino acid metabolism were enriched in the low-risk group. Among the metabolic genes, PCK1 is a gluconeogenic enzyme that has been shown to phosphorylate INSIG1/2 for lipid metabolism [95]. PCK1 was highly expressed in the low-risk group, suggesting that it may play an essential role in the tumor microenvironment.

Currently, there are insufficient research reports on PANoptosis in ccRCC patients. Our research has the advantage of establishing a good three-miRNA prognosis model based on PANoptosis, which provides the foundation and direction for future research. However, the study’s limitations must also be acknowledged. Focusing on the mRNAs associated with PANoptosis may provide a more direct and precise correlation with the PANoptosis mechanism. Due to the lack of data on effective miRNA chips and many of our clinical data for validation, this study relied solely on the data from the public database TCGA to construct the model. We attempted to explore the miRNA datasets associated with immunotherapy but came up empty, so we cannot obtain further immunotherapy data in order to validate the predictive value of the risk score. In addition, we lack the experimental conditions necessary to confirm the other possible mechanisms of these miRNAs.

In conclusion, our study demonstrated that PANoptosis is closely associated with ccRCC. The signature constructed with three PANoptosis-related miRNAs can be used as an independent prognostic molecular marker of ccRCC. In addition, its roles in immunity and therapy were thoroughly investigated. These results confirmed the clinical significance of the signature for predicting the progression of ccRCC and provided a crucial foundation for developing personalized and precise immunotherapies.

## 4. Materials and Methods

### 4.1. ccRCC Data Acquisition and Preprocessing

The expression profiles of the mRNAs and miRNAs were retrieved from The Cancer Genome Atlas (TCGA) database (https://cancergenome.nih.gov/, accessed on 20 December 2021). The samples included 71 normal kidney tissues and 545 kidney tumors, which were harvested from the TCGA-KIRC database. We downloaded the fragments per kilobase of transcript per million mapped reads (FPKM) of data for the mRNAs and raw count data for the miRNAs. For further analyses, the transcriptome data of the miRNAs were log2-transformed. The TCGA-KIRC database was also used to download the clinical information of these patients. In addition, ccRCC sample mutation data were downloaded from TCGA in VarScan2 format.

Three datasets (GSE203612, GSE74174, and GSE16441) were retrieved from GEO (http://www.ncbi.nlm.nih.gov/geo/, accessed on 9 November 2022) for further analyses and validation. The GSE203612 cell set provided single-cell RNA-seq data from 3 ccRCC samples. The “PercentageFeatureSet” calculation function, which showed a good cell viability, was used to assess the proportion of mitochondrial genes. After cell filtration, the scRNA-seq data were normalized using the LogNormalize method and the top 1500 genes with highly variable signatures were identified for further analyses. The GSE74174 dataset contained miRNA expression data and TKI effect information for 74 ccRCC patients. log2-transformation was used to normalize the raw data of the miRNAs. In the subsequent section, we calculated the risk scores using the normalized raw data and evaluated the efficacy of the TKIs. The miRNA dataset GSE16441 included 18 normal control and 18 clear renal cell carcinoma samples. The 11 major PANoptosis-related miRNAs were extracted from previous studies and are displayed in Appendix A. The selection criterion was that each miRNA had related studies for all three pathways (apoptosis, pyroptosis, and necroptosis). The final analysis did not include samples that lacked crucial clinicopathological variables or survival information.

### 4.2. Identification of Differentially Expressed miRNAs

The “limma” package in R (version 4.1.2) was used to identify the differentially expressed PRMs between the tumor and adjacent normal tissues, setting a false discovery rate (FDR) of <0.05 as the significance criteria. Pearson’s correlations among the differentially expressed miRNAs (DEMs) were calculated using the “reshape2” package. The “igraph” package was used to create the correlation networks. The DEMs in GSE16441 were computed using the GEO2R online tool (https://www.ncbi.nlm.nih.gov/geo/geo2r/, accessed on 9 November 2022), with the cutoff criteria of an adjusted *p* value < 0.05 and |log_2_FC| > 1. The DEM volcano plot was generated with https://www.bioinformatics.com.cn (accessed on 20 February 2023), an online data analysis and visualization platform.

### 4.3. Establishment of Prognostic Risk Signature and Prediction Analysis

Through the “caret” package, we randomly divided the TCGA dataset into training and testing datasets for future verification. The ratio of samples from the training group to those from the test group was 1. To screen the miRNAs related to prognosis, the previously obtained differentially expressed miRNAs were analyzed using a univariate Cox regression analysis by setting a significance threshold (*p* < 0.05) in the training group. Then, a Lasso Cox regression analysis was performed to minimize the risk of overfitting and the “glmnet” package was used to construct the prognostic features. Therefore, three miRNAs and their coefficients (β) were retained, and a PANoptosis-related prognostic signature model was established. The risk score formula of the model was as follows: Risk Score = Coef_miRNA1_ × Exp_miRNA1_ + Coef_miRNA2_ × Exp_miRNA2_ + … +Coef_miRNAi_ × Exp_miRNAi_, where Coef_miRNAi_ is the regression coefficient and Exp_miRNAi_ is the selected miRNA’s expression. According to the median risk score, the patients in the training group were divided into low-risk and high-risk groups. The two subgroups’ overall survival times were compared using a Kaplan–Meier analysis. The “prcomp” function of the “stats” R package was used to perform a principal component analysis. The “pheatmap” package plotted heatmaps to compare and visualize the miRNA expressions. The survival disparity between the two groups was represented using the “ggsurvplot” function. Finally, the 1-, 3-, and 5-year ROC curve and AUC value analysis were conducted using the “survival”, “survminer”, and “timeROC” packages. The TCGA testing dataset was utilized for validation, and the risk score was calculated using the same formula. Next, this dataset was stratified into low- and high-risk subgroups based on the median value, and the survival differences were graphically illustrated to validate the gene signature. The risk score formula was also applied to GSE74174 and the patients were divided into high- and low-risk subgroups according to the median risk score.

### 4.4. Clinical Characteristics of the Risk Score Model

The distribution of the clinicopathologic features between the risk groups was compared using heatmaps. The correlation between the clinicopathological parameters and risk attributes was evaluated using the chi-square test. In addition, a stratified survival analysis was used to test the prognostic value of our risk score model in subgroups such as age, gender, and early or late tumor grade or stage.

### 4.5. Nomogram Construction and Evaluation

Univariate and multivariate Cox regression models assessed the clinicopathological parameters influencing the survival of the ccRCC patients. We omitted the N stage due to a lack of data and retained only the remaining variables, including age, gender, grade, T/M stage, AJCC stage, and risk score. Based on the independent prognostic factors, a prognostic nomogram was generated using the “rms” package to predict the 1-year, 3-year, and 5-year OS of the ccRCC patients in the TCGA cohort. A calibration curve was drawn to evaluate the congruence between the probabilities predicted by the nomogram and the observed rates.

### 4.6. Pathway Enrichment Analysis

To investigate the potential mechanism underlying the two risk groups, GSEA was performed by selecting the c2.cp.kegg.v7.4.symbols.gmt and c5.go.v7.4.symbols.gmt annotated gene set files and setting *p* < 0.05 and FDR < 0.25 as the thresholds. The Venn diagram revealed an overlap of amino and fatty-acid-related genes and differentially expressed genes between the high- and low-risk groups. In addition, the relationship between the gene expression and risk scores was determined using Spearman’s correlation analyses.

### 4.7. Estimation of Immune Cell Infiltration and Microenvironment Characterization

The tumor microenvironment (TME) in the ccRCC samples was estimated using the Estimation of STromal and Immune cells in MAlignant Tumours using the Expression data (ESTIMATE) algorithm, which computed the stromal score, immune score, and ESTIMATE score, in addition to the tumor purity. We used various software, including XCELL, TIMER, QUANTISEQ, MCPCOUNTER, EPIC, CIBERSORT-ABS, and CIBERSORT, to thoroughly investigate the enrichment of the immune cells in the two ccRCC risk groups. Using the tools above, a bubble chart of the significantly expressed immune cell subtypes between the two risk groups was created. In addition, a single-sample gene set enrichment analysis (ssGSEA) was performed using the “gsva” package to determine the number of infiltrating immune cells and assess the activity of the immune-related pathways. The Kaplan–Meier method assessed the overall survival based on the immune cell content. The immunopheno scores (IPS) were calculated using the Sangerbox website (http://www.sangerbox.com/tool, accessed on 12 September 2022) and mRNA expression profile data from TCGA. The IPS consists of four components: antigen presentation, effector cells, suppressor cells, and checkpoints. The “maftools” package was used to compute the tumor mutational burden (TMB) value and visualize the mutation profiles of the high- and low-risk groups. In addition, we analyzed whether the immune checkpoint expressions differed between the high-risk and low-risk groups.

### 4.8. Cell Annotation and Single-Cell Trajectory Analysis

A PCA was used to eliminate the dimensions with significant separation and the tSNE algorithm was used to reduce the dimensions of the top 15 PCs, in order to obtain the main clusters. The “pheatmap” package was used to obtain the marker genes in each cluster in the heatmap, under the conditions of log2 (FC) > 2 and FDR < 0.05. The “SingleR” package then annotated the clusters into eight types of cells based on these marker genes. The “Monocle” package performed the single-cell trajectory analysis.

### 4.9. Prediction of Target Gene and Functional Analysis for miRNAs

The mRNA targeted by three candidate miRNAs was predicted using the miRDB, TargetScan, and miRTarBase databases. Only the mRNAs recognized simultaneously by all three databases were considered as candidate targets. Cytoscape (version 3.8.2, http://www.cytoscape.org/, accessed on 23 January 2022) was used to demonstrate the construction of the miRNA target gene network. Using the “cluster profiler”, “enrich plot”, “org.hs.eg.db”, and “ggplot2” packages, GO and KEGG pathway analyses were performed to investigate the biological function of the mRNAs targeted by the miRNAs.

### 4.10. Drug Sensitivity Analysis

We chose Genomics of Drug Sensitivity in Cancer (GDSC, https://www.cancerrxgene.org, accessed on 18 August 2022) to compare the responses to chemotherapeutic and small-molecule drugs between the high- and low-risk groups. The IC50 displayed the drug sensitivity results. The analysis used the “pRRophetic” and “oncopredict” packages.

### 4.11. Clinical Samples

Twenty-six paired clinical clear-cell renal cell carcinoma samples and adjacent normal samples were obtained with informed consent from Sir Run Run Shaw Hospital, School of Medicine, Zhejiang University, between 2013 and 2019. Before resection, none of the patients had received chemotherapy or radiotherapy. The Ethics Committees of Sir Run Run Shaw Hospital, the School of Medicine, Zhejiang University, approved this study. Written informed consent was obtained from the patients to publish this paper. 

### 4.12. Cell Culture

Two ccRCC cell lines 786-O and Caki-1 were acquired from the Institute of Biochemistry and Cell Biology of the Chinese Academy of Sciences (Shanghai, China). 786-O was maintained in RPMI-1640 medium (Invitrogen, 11875093, Shanghai, China) supplemented with 10% fetal calf serum (Gibco, 12483012, Beijing, China). The Caki-1 cell line was grown in McCoy 5A (Invitrogen, 16600082, Beijing, China) medium containing 10% fetal calf serum. The two cell lines were both incubated in a humidified incubator with 95% air and 5% CO_2_.

### 4.13. Cell Transfection

Gene Pharma Company (Shanghai, China) designed and synthesized all the mimics, inhibitors, and corresponding negative controls. For the transfections, the cells were seeded overnight and transfected with Lipofectamine^TM^ RNAiMAX transfection reagent (Thermo Fisher Scientific, 13778500, Carlsbad, CA, USA), as per the manufacturer’s instructions. The sequences of the mimics and inhibitors used in this study are listed in Appendix A.

### 4.14. RNA Extraction and Real-Time Quantitative PCR

TRIzol™ Reagent (Invitrogen, 15596026, Shanghai, China) was used to extract the total RNA from the tissue samples and NanoDrop 2000 (Wilmington, DE, USA) was used to measure the concentrations. The miRNA extraction from the cells was performed using the miRNeasy Mini Kit (QIAGEN, 217004, Shanghai, China). The miRNeasy Mini Kit (QIAGEN, 217004, Shanghai, China) was used to extract the miRNA from the cells. The miRNA First Strand cDNA Synthesis (Tailing Reaction) Kit (Sangon Biotech, B532451, Shanghai, China) was used to reverse-transcribe the miRNAs, following the manufacturer’s instructions. The SYBR Green Master Mix Kit and Light Cycler 480 II systems (Roche, Shanghai, China) were used for the real-time quantitative PCR (RT-qPCR) to determine the miRNA expression. For normalization, U6 small RNA levels were used. The relative expressions of the miRNAs were determined using the 2^−∆∆Ct^ method. The study’s forward primers are listed in Appendix A and the kit supplied the universal downstream primers.

### 4.15. Cell Counting Kit-8 Assay

The Cell Counting Kit-8 (CCK-8) assay kit (YEASEN, 40203ES80, Shanghai, China) determined the viable cell mass. After 48 h of transfection with mimics or inhibitors, the 786-O cells (5 × 10^3^) were seeded in 96-well plates. The cells were then cultured for three days at 37 °C in an incubator with 5% CO_2_. Each well received 10 µL of CCK-8 solution, and the cells were cultured for an additional 2 h. The absorbance was finally measured at 450 nm using a microplate reader. 

### 4.16. Statistical Analysis

All the statistical analyses in this study used R (version 4.1.2). The Student’s t-test and Wilcoxon test were used to assess the statistical significance of the normally and non-normally distributed quantitative variables. We evaluated the survival data using the Kaplan–Meier method and a two-sided log-rank test. We evaluated the hazard ratio (HR) and 95% confidence interval (CI) in univariate and multivariate Cox regression models. *p* < 0.05 indicated a statistically significant difference in the data (* *p* < 0.05, ** *p* < 0.01, and *** *p* < 0.001).

## 5. Conclusions

The current study screened PRMs and constructed a prognostic risk model of three miRNAs, which revealed a favorable prognostic value for ccRCC and validated the biological behaviors of miRNAs.

Using multiple datasets, we conducted a comprehensive immunogenomic analysis to demonstrate the high-risk group’s immunological profile and potential immune escape mechanisms based on the prognostic risk model.

The novel PANoptosis-based score can be a reliable predictor of prognosis and immune response and may offer a novel treatment strategy for ccRCC.

## Figures and Tables

**Figure 1 ijms-24-09392-f001:**
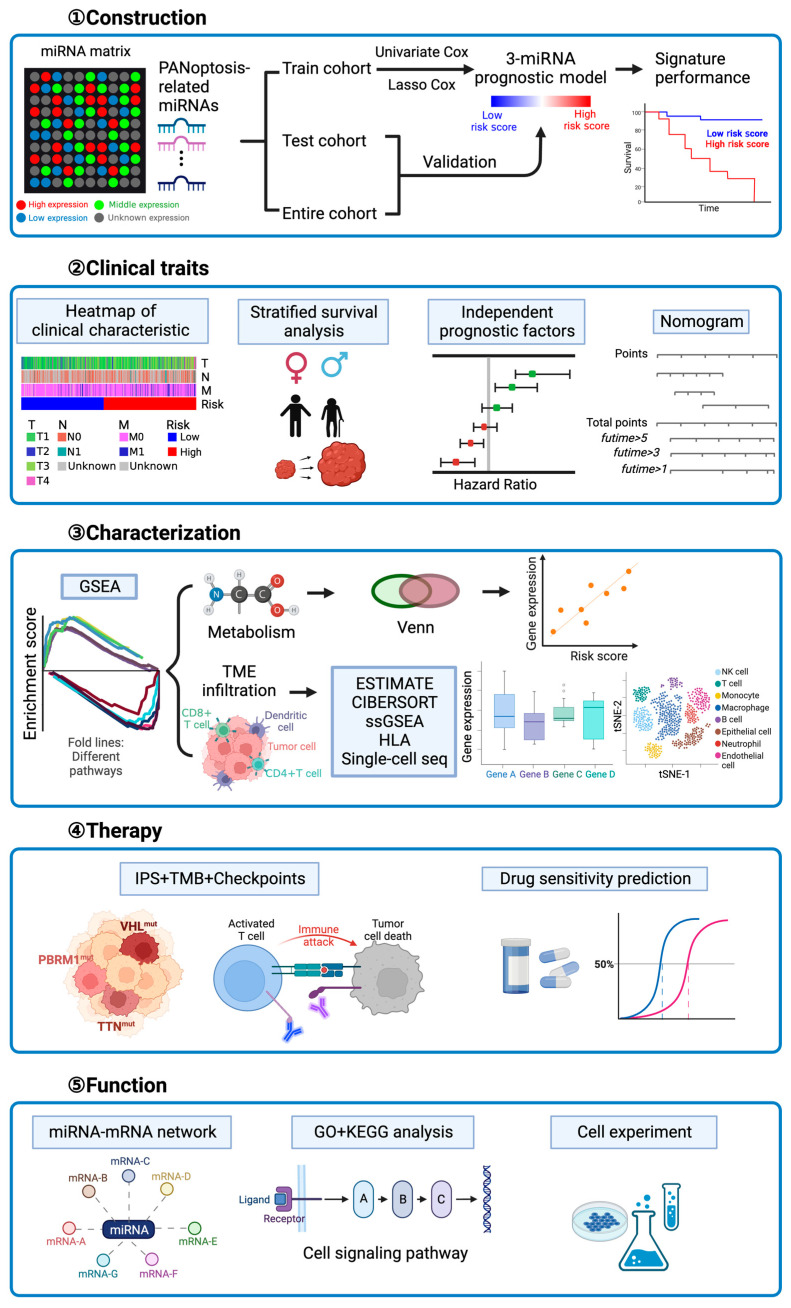
Flowchart of the study. Step 1: Construction of the PANoptosis-related miRNA prognostic model; Step 2: Exploration of the clinical traits of the prognostic model; Step 3: Characterization of the high- and low-risk groups; Step 4: Estimation of the immunotherapy and chemotherapy responses; and Step 5: Functional enrichment analysis of miRNA. This figure was created with BioRender.com.

**Figure 2 ijms-24-09392-f002:**
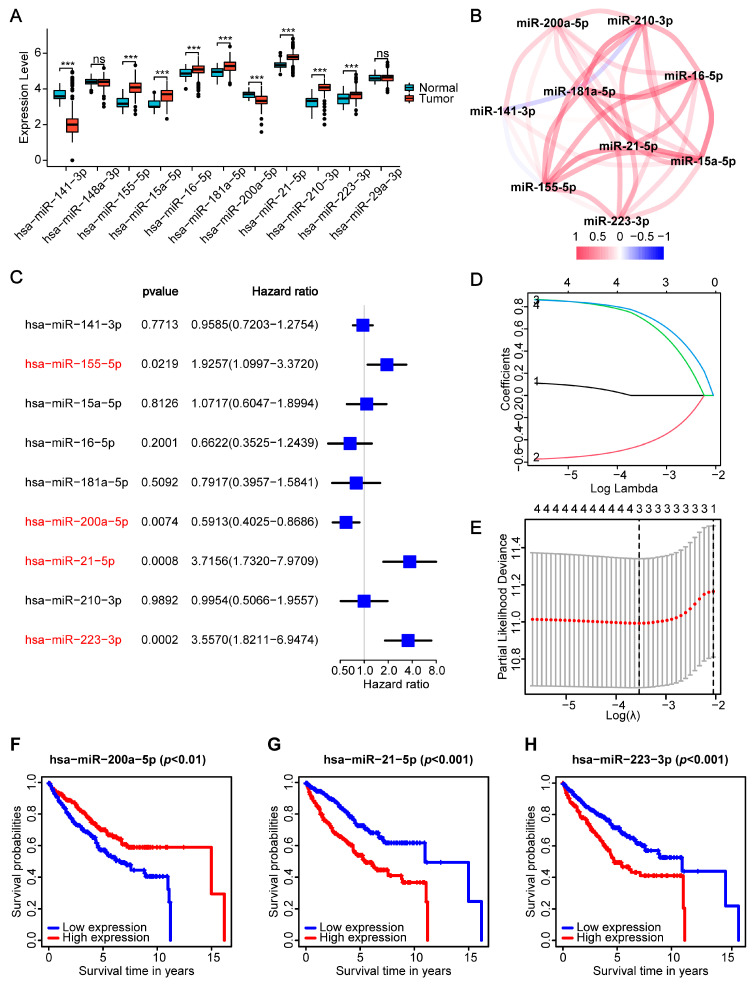
Construction of the prognostic risk signature based on PRMs for ccRCC. (**A**) The expression of 11 PRMs in tumor and adjacent normal tissues (PRMs, PANoptosis-related microRNAs). *** FDR < 0.001, ns: not significant (FDR, false discovery rate). (**B**) The correlation network of the nine differentially expressed PRMs (red line: positive correlation; blue line, negative correlation. The intensity of the colors reflects the level of significance). (**C**) Univariate Cox regression analysis of OS for PRMs (OS, overall survival) and four miRNAs with *p* < 0.05 marked in red. (**D**) Lasso Cox regression of the four PRMs. Each curve represents the change trajectory of each gene coefficient with the log lambda sequence. (**E**) Cross-validation for tuning the parameter selection in the Lasso regression. The red dashed line refers to the variation curve of partial likelihood deviation with log lambda. The gray line refers to the number of variables included under the corresponding log lambda. The black vertical dashed lines were drawn at the optimal values by using the minimum criteria (left line) and the 1 standard error of the minimum criteria (right line). (**F**–**H**) The Kaplan–Meier analysis of the OS for the three prognostic PRMs.

**Figure 3 ijms-24-09392-f003:**
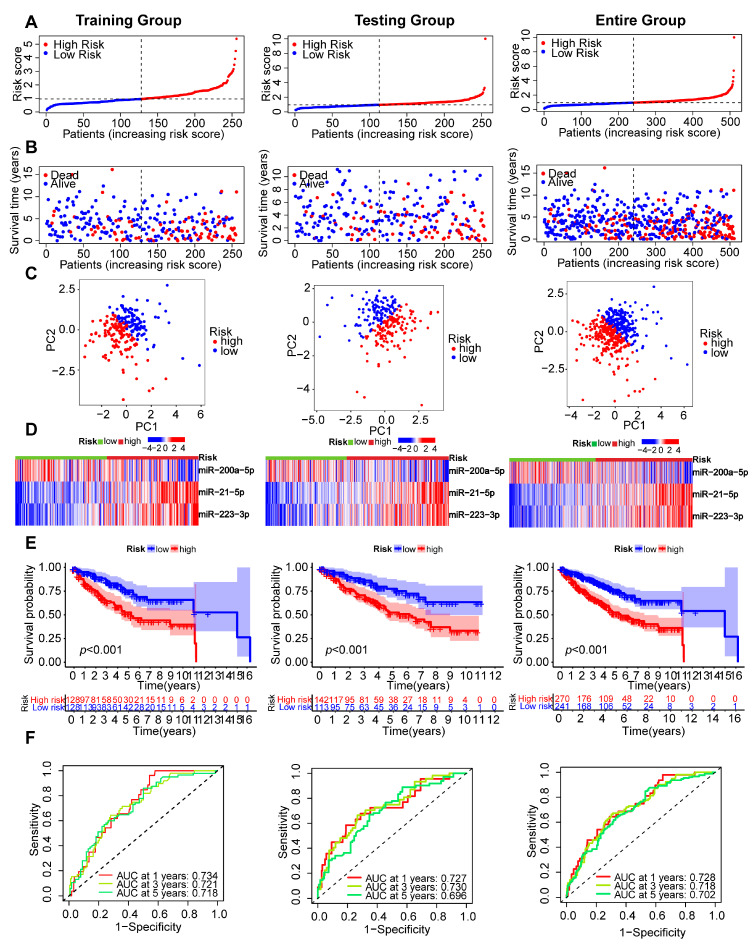
Assessment and validation of the PANoptosis-related prognostic model in testing, training, and entire sets. (**A**,**B**) The distribution of risk scores and survival status of ccRCC patients. (**C**) The PCA plot of ccRCC patients (PCA, principal component analysis). (**D**) A heatmap depicting the expressions of the three prognostic PRMs between high- and low-risk groups. (**E**) The OS analysis. (**F**) The time-dependent ROC curve analysis (ROC, receiver operating characteristic).

**Figure 4 ijms-24-09392-f004:**
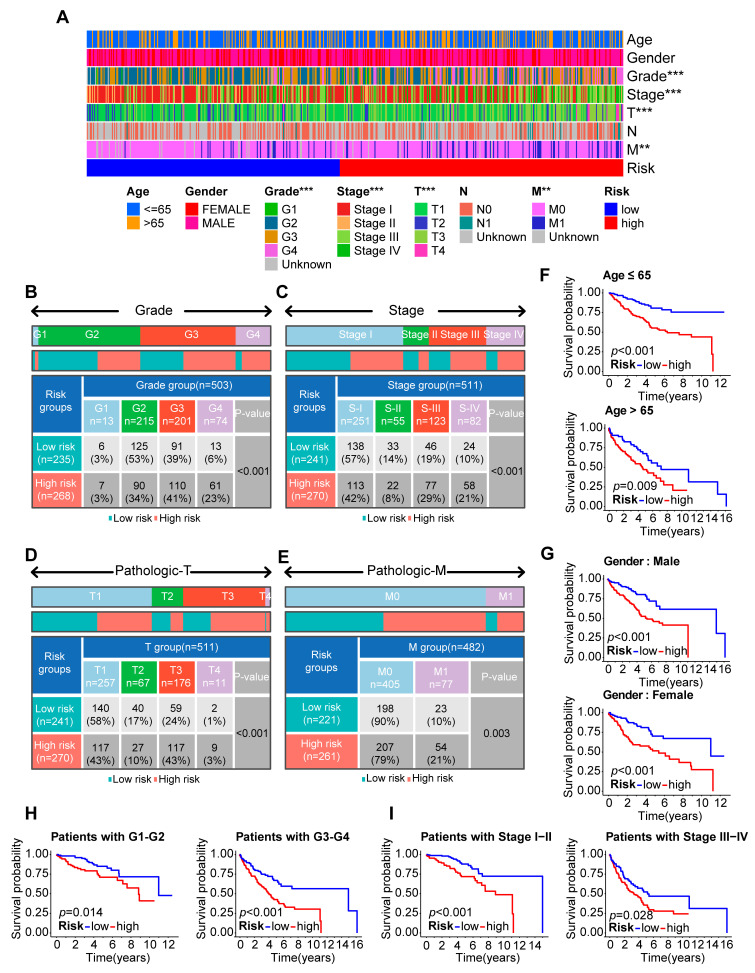
The relationship between various clinical characteristics and the PANoptosis-related prognostic model. (**A**) A heatmap that showed the distribution of clinicopathologic characteristics between the high- and low-risk groups. The correlation between clinicopathological parameters and risk attributes (high or low) was evaluated using a chi-square test. ** *p* < 0.01, *** *p* < 0.001. (**B**–**E**) The specific chi-square tests for the grade, AJCC stage, T stage, and M stage. (**F**) The survival difference was stratified by age (age ≤ 65, age > 65). (**G**) The survival difference was stratified by gender (male, female). (**H**) The survival difference was stratified by grade (G1–G2, G3–G4). (**I**) The survival difference was stratified by AJCC stage (Stage I–II, Stage III–IV).

**Figure 5 ijms-24-09392-f005:**
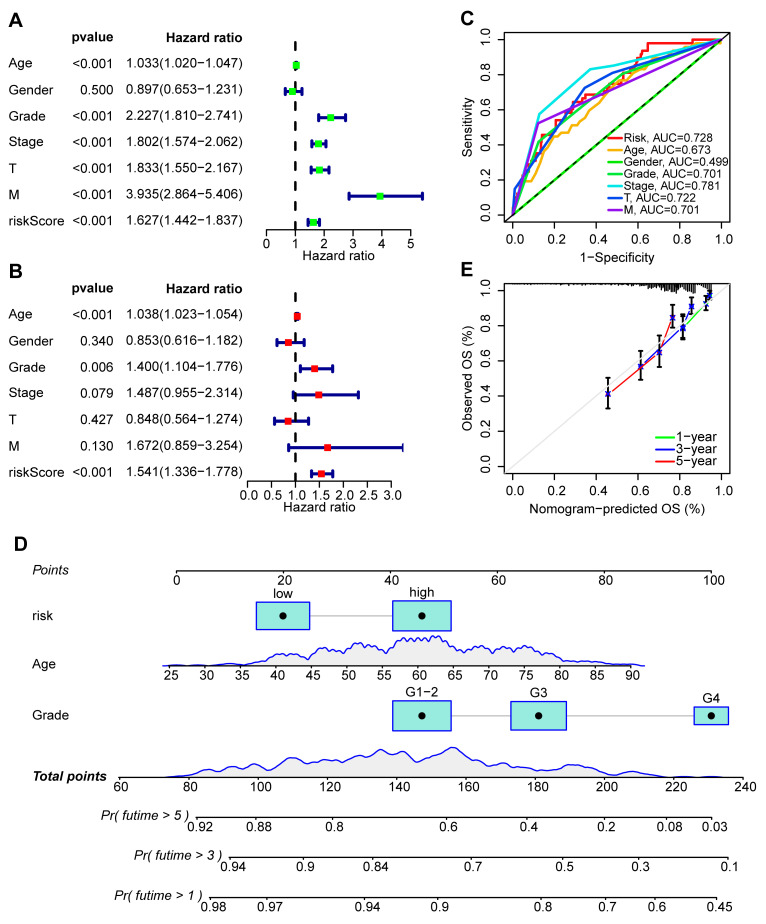
Nomogram construction and evaluation. (**A**) Univariate Cox regression using risk scores and other clinical characteristics. (**B**) Multivariate Cox regression demonstrated that age, grade, and risk score were independent prognostic factors for OS in ccRCC patients. (**C**) An ROC curve illustrating the prognostic accuracy of various clinical characteristics. (**D**) The nomogram was constructed using age, grade, and risk score to predict the 1-, 3-, and 5-year OS of ccRCC patients. (**E**) The calibration curve of the nomogram.

**Figure 6 ijms-24-09392-f006:**
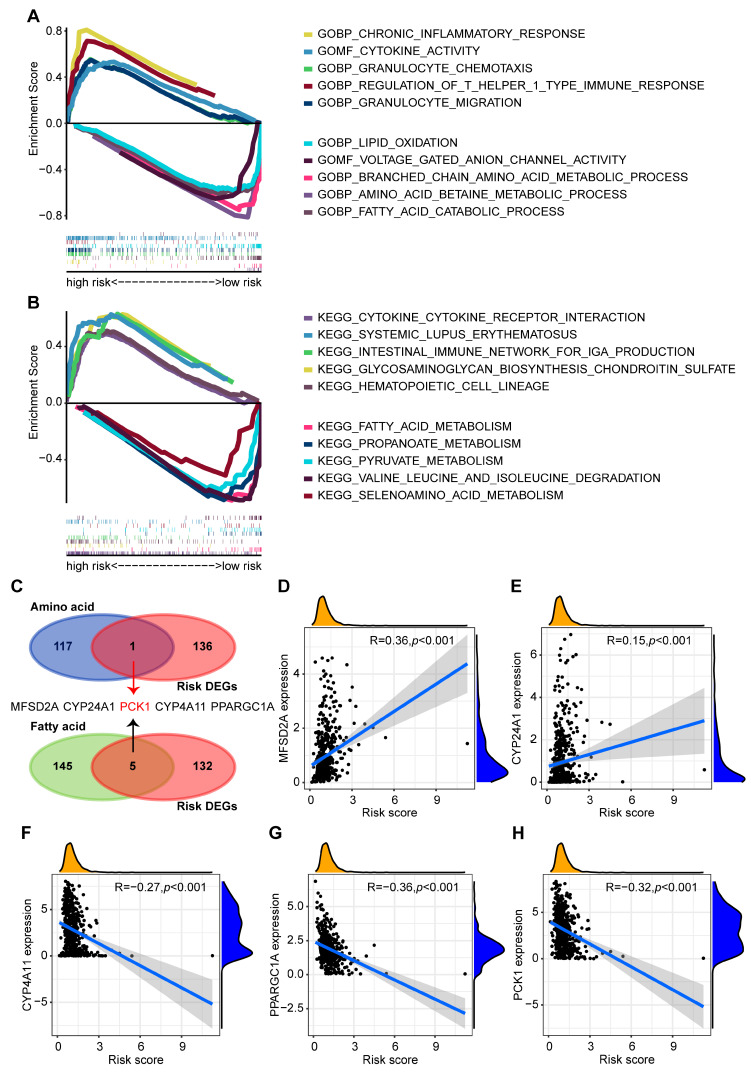
Functional analysis between the two groups. (**A**) GSEA was based on GO between the high- and low-risk groups (GSEA, gene set enrichment analysis; GO, gene ontology). (**B**) GSEA was based on KEGG between the high- and low-risk groups (KEGG, Kyoto Encyclopedia of Genes and Genomes). (**C**) The Venn diagram revealed the overlap of amino acid/fatty acid-related genes and differentially expressed genes (DEGs) between the high-risk and low-risk groups. The PCK1 gene marked in red was involved in both lipid and amino acid metabolism. (**D**–**H**) The correlation analysis between five identified genes and risk score. The black dots represent samples and blue lines represent regression lines for correlations.

**Figure 7 ijms-24-09392-f007:**
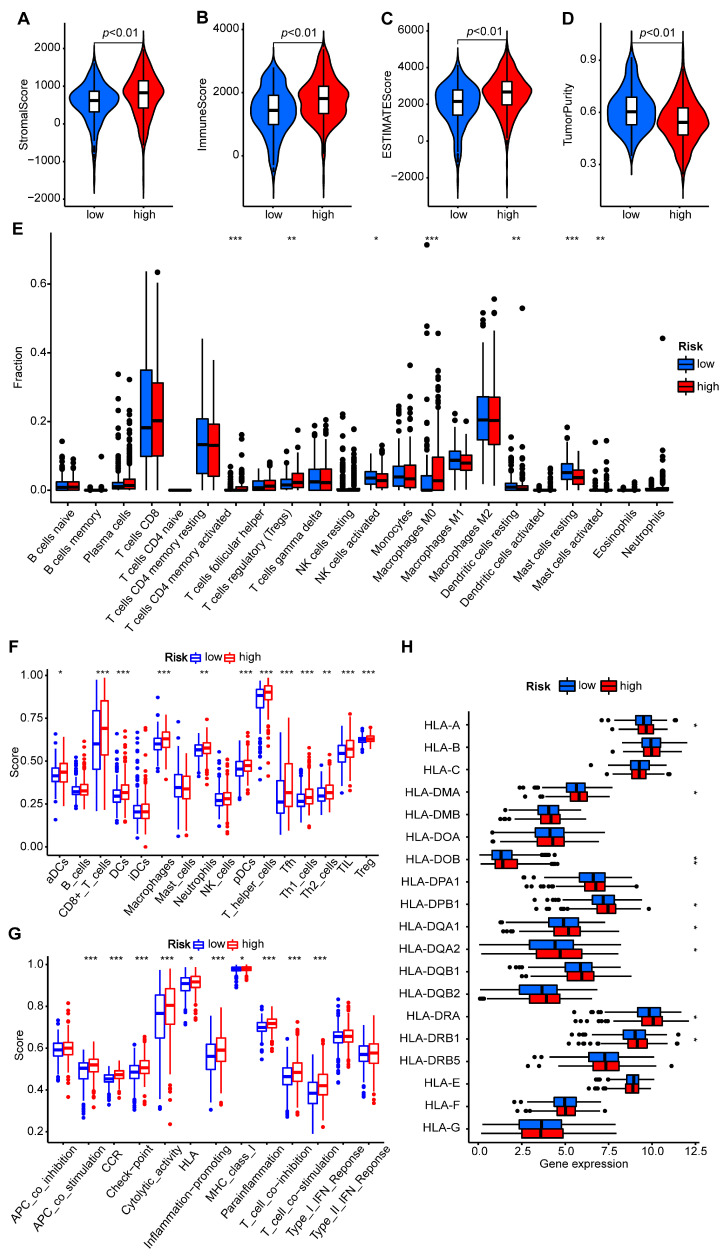
Immune cell infiltration and microenvironment characterization of the high- and low-risk groups. (**A**–**D**) The stromal, immune, and ESTIMATE scores and tumor purity between the two groups. (**E**) The abundances of 22 types of immune cells determined by the CIBERSORT analytical tool for different cell groups. (**F**,**G**) A comparison of the enrichment scores of 16 types of immune cells and 13 immune-related pathways using ssGSEA. (**H**) The expression of HLA gene sets between the high- and low-risk cell groups. * *p* < 0.05, ** *p* < 0.01, and *** *p* < 0.001.

**Figure 8 ijms-24-09392-f008:**
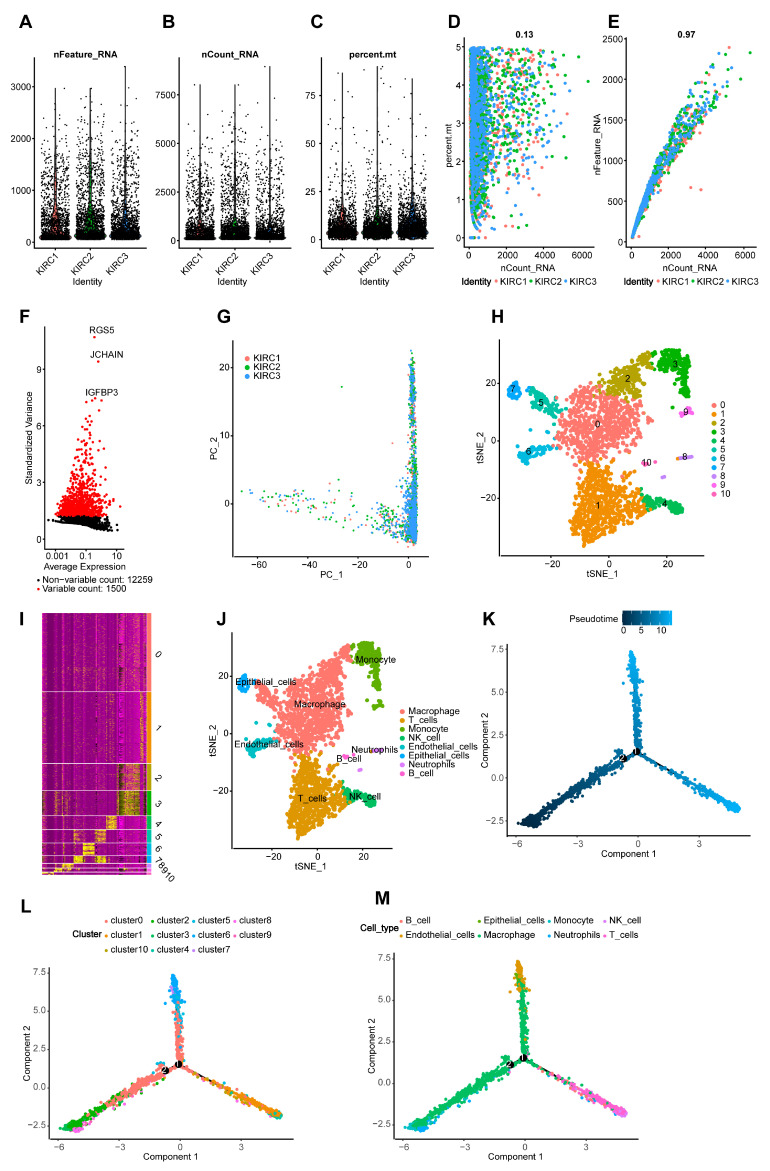
Quality control and filtration of single-cell RNA-seq data, annotation of cell types, and single-cell trajectory analysis. (**A**) The number of genes detected in 3 ccRCC samples. (**B**) The sequencing depth of each sample. (**C**) The proportion of mitochondrial genes in each sample. (**D**) There is a positive correlation between sequencing depth and mitochondrial gene content in samples (R = 0.13). (**E**) There is a positive correlation between sequencing depth and total intracellular sequences in samples (R = 0.97). (**F**) The volcano plot revealed fluctuating genes in all samples. Red dots indicate the top 1500 genes; the top three are listed. (**G**) A preliminary dimensionality reduction PCA plot of scRNA-seq data (scRNA-seq, single-cell RNA-seq). (**H**) The clustering of ccRCC cells was performed using tSNE. (**I**) The heatmap of marker genes in each cluster. Genes upregulated and downregulated are represented by yellow and purple, respectively. (**J**) The cell type annotation of clusters. According to marker genes, 11 clusters were divided into eight cell types. (**K**) Single-cell pseudotime analysis of three subsets. (**L**,**M**) Trajectory analysis of clusters and cell types. Different colored dots were placed on pseudotime branches to represent clusters or cell types.

**Figure 9 ijms-24-09392-f009:**
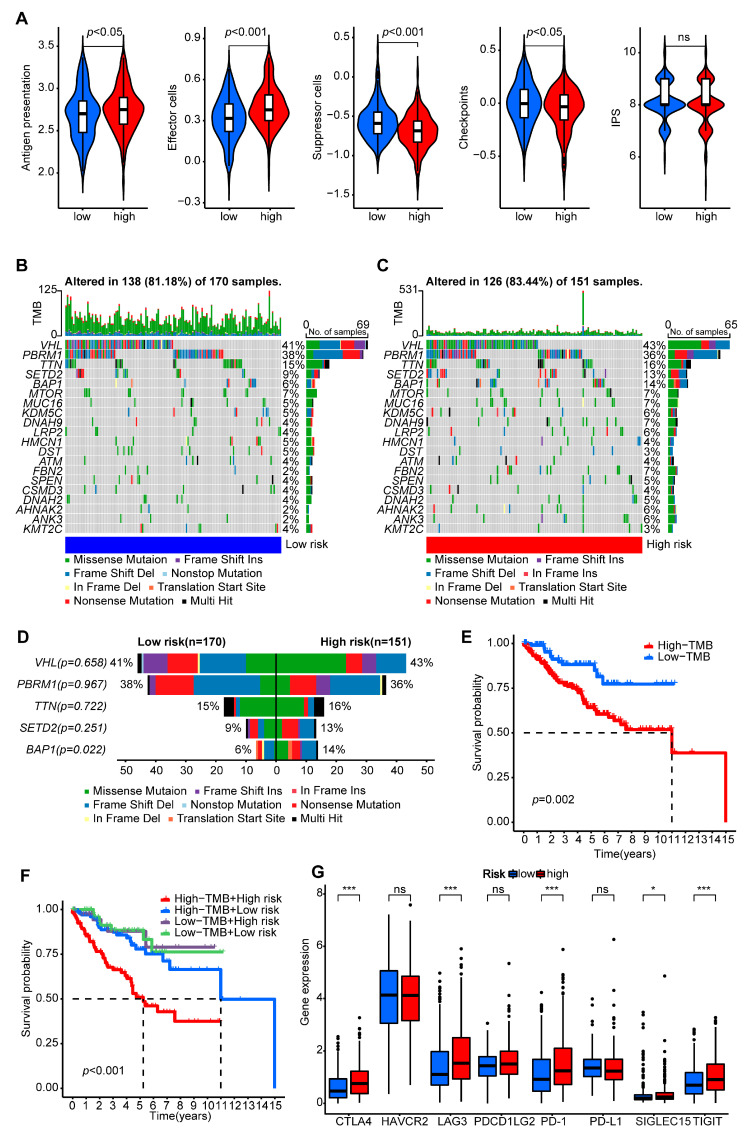
Immunotherapy prediction based on the signature model. (**A**) The antigen presentation, effector cells, suppressor cells, checkpoints, and IPS were determined. (**B**,**C**) The mutation profiles of the high- and low-risk groups. (**D**) A chi-square test comparison of the mutation rate between the high- and low-risk groups. (**E**) The OS of high-TMB and low-TMB using Kaplan–Meier in the log-rank test. (**F**) The OS of the patients was stratified by both the risk score and TMB using Kaplan–Meier curves. (**G**) The expression of immune checkpoint molecules between two groups. * *p* < 0.05, *** *p* < 0.001, and ns: not significant.

**Figure 10 ijms-24-09392-f010:**
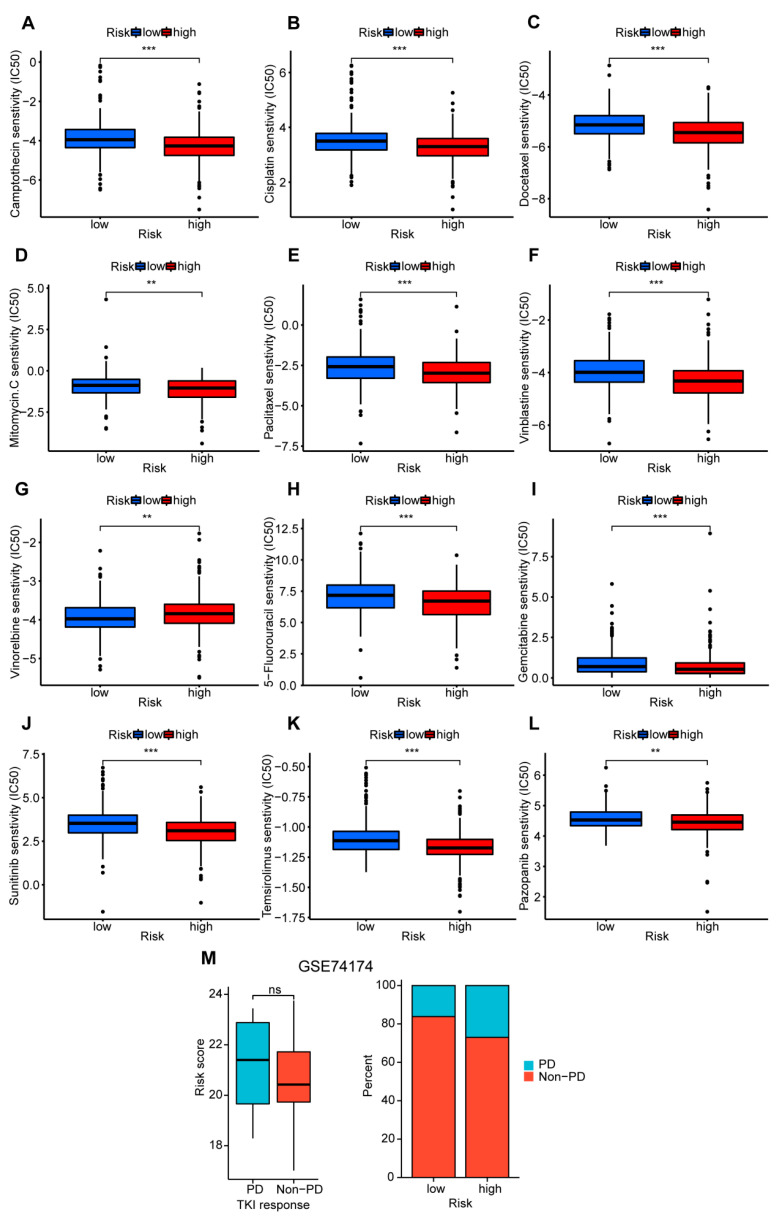
Anticancer drugs analysis. (**A**–**L**) Drug sensitivity analysis for the high- and low-risk groups. (**M**) Box plots and bar graphs depicting the therapy response to TKI treatment in the high- and low-risk subtypes of the GSE74174 dataset (TKI, tyrosine kinase inhibitor). ** *p* < 0.01, *** *p* < 0.001, and ns: not significant.

**Figure 11 ijms-24-09392-f011:**
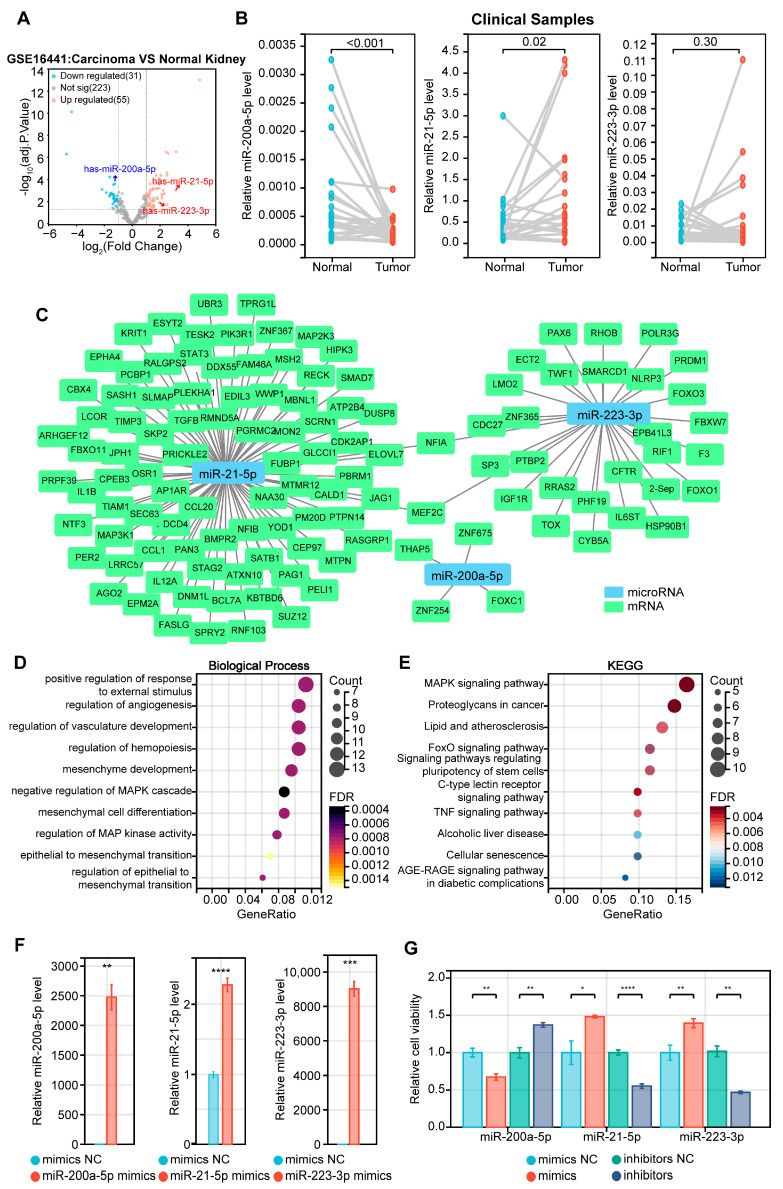
Functional annotation analysis of mRNAs in the miRNA–mRNA network. (**A**) A volcano plot depicting differentially expressed miRNAs in GSE16441. The red dot represents miRNAs that are upregulated and the blue dot represents miRNAs that are downregulated. (**B**) Relative expression levels of three prognostic PRMs in clinically paired kidney tumors and adjacent normal tissue. (**C**) Construction of a miRNA–mRNA co-expression network based on 3 PRMs and their highly correlated genes. (**D**) The biological process (BP) result of the GO analysis of the targeted mRNAs. (**E**) KEGG enrichment analysis of the targeted mRNAs. (**F**) The transfection efficiency of miRNAs mimics in 786-O cells. (**G**) Cell Counting Kit-8 (CCK-8) assay of 786-O cell viability changes with miRNA mimics and inhibitors transfection. * *p* < 0.05, ** *p* < 0.01, *** *p* < 0.001 and **** *p* < 0.0001.

**Table 1 ijms-24-09392-t001:** Characteristics of patients with ccRCC included in this study.

Characteristics	Variable	Total (*n* = 511)	Train Cohort (*n* = 256)	Test Cohort (*n* = 255)	Chi-Square (*p*)
Age	≤65	338 (66.14%)	170 (66.41%)	168 (65.88%)	0.9004
	>65	173 (33.86%)	86 (33.59%)	87 (34.12%)	
Gender	Male	332 (64.97%)	169 (66.02%)	163 (63.92%)	0.6198
	Female	179 (35.03%)	87 (33.98%)	92 (36.08%)	
Grade	G1	13 (2.54%)	8 (3.13%)	5 (1.96%)	0.202
	G2	215 (42.07%)	110 (42.97%)	105 (41.18%)	
	G3	201 (39.33%)	95 (37.11%)	106 (41.57%)	
	G4	74 (14.48%)	36 (14.06%)	38 (14.9%)	
	Unknown	8 (1.57%)	7 (2.73%)	1 (0.39%)	
Pathologic stage	Stage I	251 (49.12%)	125 (48.83%)	126 (49.41%)	0.9046
	Stage II	55 (10.76%)	29 (11.33%)	26 (10.2%)	
	Stage III	123 (24.07%)	59 (23.05%)	64 (25.1%)	
	Stage IV	82 (16.05%)	43 (16.8%)	39 (15.29%)	
Tumor (T)	T1	257 (50.29%)	128 (50%)	129 (50.59%)	0.8356
	T2	67 (13.11%)	34 (13.28%)	33 (12.94%)	
	T3	176 (34.44%)	87 (33.98%)	89 (34.9%)	
	T4	11 (2.15%)	7 (2.73%)	4 (1.57%)	
Node (N)	N0	227 (44.42%)	107 (41.8%)	120 (47.06%)	0.1988
	N1	16 (3.13%)	11 (4.3%)	5 (1.96%)	
	Unknown	268 (52.45%)	138 (53.91%)	130 (50.98%)	
Metastasis (M)	M0	405 (79.26%)	202 (78.91%)	203 (79.61%)	0.4726
	M1	77 (15.07%)	42 (16.41%)	35 (13.73%)	
	Unknown	29 (5.68%)	12 (4.69%)	17 (6.67%)	
Survival status	Alive	340 (66.54%)	170 (66.41%)	170 (66.67%)	0.6576
	Dead	171 (33.46%)	86 (33.59%)	85 (33.33%)	

## Data Availability

The datasets analyzed for this study can be found in TCGA (https://portal.gdc.cancer.gov/, accessed on 9 November 2022) and GEO (https://www.ncbi.nlm.nih.gov/geo/, accessed on 9 November 2022). All data relevant to the experiments are included in the Appendix A.

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
