# Peer review of "A Novel Defined PANoptosis-Related miRNA Signature for Predicting the Prognosis and Immune Characteristics in Clear Cell Renal Cell Carcinoma: A miRNA Signature for the Prognosis of ccRCC"

_ijms, 2023, doi:10.3390/ijms24119392_

Round 1

Reviewer 1 Report

The authors constructed a PANoptosis-related microRNA signature and revealed its potential significance on the clinicopathological features and clear cell renal cell carcinoma (ccRCC) immunity. According to the authors, the control of the inflammatory response is one of the potential ways to suppress tumors providing new precise treatment strategies in the era of precision medicine (doi: 10.21037/jtd.2017.08.68).

The work is well developed, but some revisions are needed, in my opinion:

- Kidney Renal clear cell carcinoma is composed of double same-meaning words: Kidney and Renal. In the scientific literature, we talk of clear cell renal cell carcinoma (ccRCC). Please modify the extension in the title and short form in the manuscript.

- Please modify the running title as ccRCC.

- In the Introduction section, please be clearer on what is the best treatment for metastatic ccRCC (doi: 10.1016/j.euo.2019.09.002)

- In the Introduction section, please improve knowledge on miRNA and renal cell carcinoma (doi: 10.3390/cancers14051112)

- Please report the Materials and Methods section before the Results section

- Please also improve knowledge on the importance of miRNA in the diagnostic algorithm (doi: 10.1016/j.urolonc.2021.11.001)

The English language is quite fine, with only minor mistakes.

Reviewer 2 Report

This is a novel area of research which indicates assessment of miRNAs that regulate PANoptosis may be clinically useful. While preliminary in nature, the novelty makes this study of significant interest. Figure 1 is very helpful. A major concern is that how the high- and low-risk groups were generated is poorly described. I have several other concerns and suggestions that are listed below.

1.       Line: 55-56: Please include citations for review papers which provide an overview of PANoptosis and support this statement.

2.       Line 95-96: Please provide a citation.

3.       Methods section: How patients were divided into high- and low-risk groups need to be more clearly explained. Lines 531-533 state: ‘In the following section, we computed the risk score using normalized raw count data and analyzed the effectiveness of the TKIs’ – I don’t see any mention of this in the next section (section 4.3). Did this get left out by accident? Perhaps line 560 needs to be expanded? As is, this is very unclear and needs to be explained.

4.       Results section: I think some details regarding how patients were divided into high- and low-risk groups also need to be included in the results section to help give context to the results being described.

5.       Figure 3A: typo – says ‘socre’ instead of ‘score’

6.       Inclusion of only one cell line is a major weakness, typically data needs to be collected from at least 2 cell lines to make inferences based on the data generated. My suggestion is to frame these data as pilot data and note that studies with additional cell lines will be needed to make conclusions (lines 376-381) and discussion section.
